# Mental Health and Substance Use Associated with Hospitalization among People with COVID-19: A Population-Based Cohort Study

**DOI:** 10.3390/v13112196

**Published:** 2021-10-31

**Authors:** Héctor Alexander Velásquez García, James Wilton, Kate Smolina, Mei Chong, Drona Rasali, Michael Otterstatter, Caren Rose, Natalie Prystajecky, Samara David, Eleni Galanis, Geoffrey McKee, Mel Krajden, Naveed Zafar Janjua

**Affiliations:** 1British Columbia Centre for Disease Control, Vancouver, BC V5Z 4R4, Canada; hector.velasquez@bccdc.ca (H.A.V.G.); james.wilton@bccdc.ca (J.W.); kate.smolina@bccdc.ca (K.S.); mei.chong@bccdc.ca (M.C.); drona.rasali@bccdc.ca (D.R.); michael.otterstatter@bccdc.ca (M.O.); caren.rose@bccdc.ca (C.R.); natalie.prystajecky@bccdc.ca (N.P.); samara.david@phac-aspc.gc.ca (S.D.); eleni.galanis@bccdc.ca (E.G.); geoffrey.mckee@bccdc.ca (G.M.); mel.krajden@bccdc.ca (M.K.); 2School of Population and Public Health, University of British Columbia, Vancouver, BC V6T 1Z3, Canada; 3Department of Pathology and Laboratory Medicine, University of British Columbia, Vancouver, BC V6T 1Z7, Canada; 4Centre for Health Evaluation and Outcome Sciences, St. Paul’s Hospital, Vancouver, BC V6Z 1Y6, Canada

**Keywords:** COVID-19, cohort studies, registries, risk factors, hospitalization, mental health, substance-related disorders, diabetes mellitus, intellectual disability, pregnancy

## Abstract

This study identified factors associated with hospital admission among people with laboratory-diagnosed COVID-19 cases in British Columbia. The study used data from the BC COVID-19 Cohort, which integrates data on all COVID-19 cases with data on hospitalizations, medical visits, emergency room visits, prescription drugs, chronic conditions and deaths. The analysis included all laboratory-diagnosed COVID-19 cases in British Columbia to 15 January 2021. We evaluated factors associated with hospital admission using multivariable Poisson regression analysis with robust error variance. Of the 56,874 COVID-19 cases included in the analysis, 2298 were hospitalized. Factors associated with increased hospitalization risk were as follows: male sex (adjusted risk ratio (aRR) = 1.27; 95% CI = 1.17–1.37), older age (*p*-trend < 0.0001 across age groups increasing hospitalization risk with increasing age [aRR 30–39 years = 3.06; 95% CI = 2.32–4.03, to aRR 80+ years = 43.68; 95% CI = 33.41–57.10 compared to 20–29 years-old]), asthma (aRR = 1.15; 95% CI = 1.04–1.26), cancer (aRR = 1.19; 95% CI = 1.09–1.29), chronic kidney disease (aRR = 1.32; 95% CI = 1.19–1.47), diabetes (treated without insulin aRR = 1.13; 95% CI = 1.03–1.25, requiring insulin aRR = 5.05; 95% CI = 4.43–5.76), hypertension (aRR = 1.19; 95% CI = 1.08–1.31), injection drug use (aRR = 2.51; 95% CI = 2.14–2.95), intellectual and developmental disabilities (aRR = 1.67; 95% CI = 1.05–2.66), problematic alcohol use (aRR = 1.63; 95% CI = 1.43–1.85), immunosuppression (aRR = 1.29; 95% CI = 1.09–1.53), and schizophrenia and psychotic disorders (aRR = 1.49; 95% CI = 1.23–1.82). In an analysis restricted to women of reproductive age, pregnancy (aRR = 2.69; 95% CI = 1.42–5.07) was associated with increased risk of hospital admission. Older age, male sex, substance use, intellectual and developmental disability, chronic comorbidities, and pregnancy increase the risk of COVID-19-related hospitalization.

## 1. Introduction

COVID-19 caused by SARS-CoV-2, has affected millions of people globally and can generate a spectrum of health outcomes among those infected. Clinical presentation can range from asymptomatic/mild illnesses to severe disease that requires hospitalization and intensive care [1,2,3].

The overall goal of the pandemic response is to minimize severe disease, overall deaths and societal disruption. Various demographic factors (e.g., older age, male sex) and chronic comorbidities (e.g., diabetes, cardiovascular disease (CVD), hypertension) have been identified as risk factors for hospitalization and other severe outcomes [2]. However, most studies have been conducted on patients presenting at hospitals, and there are few population-based studies evaluating risk factors in all COVID-19-diagnosed individuals in a specific jurisdiction. Limiting analyses to hospitalized patients may lead to potential selection bias when characterizing risk factors. Furthermore, even though older age has been identified as the strongest risk factor for severe disease along with various comorbidities [4], very few studies have investigated the relationship of substance use, intellectual disabilities and insulin-dependent diabetes with the risk of severe outcomes [5,6].

Identification of risk factors for COVID-19 hospitalization is important for the prioritization of interventions aimed at reducing health system burden and maintaining hospital capacity. However, risk factors may differ by jurisdiction as the evolution of the COVID-19 pandemic has not been uniform globally. Further, the clinical threshold for hospital admission may vary across settings, especially early in the pandemic. In this study, we identified factors associated with hospital admission among people with COVID-19 infection in British Columbia (BC).

## 2. Materials and Methods

### 2.1. Study Population

This study used data from the BC COVID-19 Cohort (BCC19C), which integrates data on all individuals tested for COVID-19 in BC, with data on COVID-19 hospital and ICU admissions, medical visits, other hospitalizations, emergency room visits, prescription drugs, chronic conditions and mortality (Appendix A). The BCC19C was established as a public health surveillance system under the BCCDC’s public health mandate. This study was reviewed and approved by the Behavioural Research Ethics Board at the University of British Columbia (approval # H20-02097).

The study population for this analysis included individuals who tested positive for SARS-CoV-2 by real-time reverse transcription–polymerase chain reaction (RT-PCR), from 26 January 2020 to 15 January 2021. The outcome of interest was hospitalization (as a measure of COVID-19 severity), defined as hospital admission in a BC acute care facility within 14 days after a positive SARS-CoV-2 test [7,8,9]. Patients residing in long term care facilities were excluded from the analyses as their transfer to hospitals was variable over time and across local regions. For women of reproductive age (15–49 years), hospital admissions were only considered related to COVID-19 if no obstetric-related hospitalization codes were found in the Discharge Abstract Database (DAD) within 14 days of hospital admission in Appendix A.

### 2.2. Comorbidities

We examined the following chronic conditions: Alzheimer/dementia, asthma, chronic heart disease (CHD: acute myocardial infarct, angina, heart failure, ischemic myocardial infarct), chronic obstructive pulmonary disease (COPD), cirrhosis, chronic kidney disease (CKD), depression, diabetes (categorized as no-diabetes, treated without insulin and requiring insulin), epilepsy, gout, hypertension, stroke (ischemic, haemorrhagic, transitory ischemic attack), mood and anxiety disorders, osteoarthritis, osteoporosis, parkinsonism, rheumatoid arthritis, substance use disorder, injection drug use (IDU), problematic alcohol use, cancer, immunosuppression, intellectual and developmental disabilities (IDD) and schizophrenia and psychotic disorders (SZP). Variable definitions and diagnostic codes used to identify comorbidities are detailed in Appendix A.

### 2.3. Statistical Analysis

We described the baseline characteristics of participants including age, sex and pregnancy status. We evaluated risk factors associated with hospital admission, calculating risk ratios through multivariable Poisson regression models with robust error variance [10]. Analyses were conducted treating age as continuous as well as categorized into groups. Model-building started with a model including age and sex, followed by other variables. Added variables were evaluated through Wald’s tests and improvement of model fit was determined through the Akaike Information Criterion. To assess population differences across time, the cohort was stratified according to two waves or time periods: 26 January to 1 August 2020 and 2 August 2020 to 15 January 2021. Sensitivity analyses were performed by (1) stratifying the population by age group and (2) by restricting the outcome to hospitalizations lasting more than two days to address severity. All statistical analyses were performed using R version 4.0.2 [11].

## 3. Results

### 3.1. Demographic and Clinical Characteristics

The analysis included 56,874 COVID-19 cases diagnosed before 15 January 2021, 2298 (4.0%) of whom were hospitalized. Males represented 51.2% of the people reported as COVID-19 positive and a greater percentage of hospital admissions (58.5%) (Table 1). The median age of COVID-19 cases was 35 years (IQR: 24–50), while that of people requiring hospital admission was nearly twice that (66 years; IQR: 53–78). The hospitalization rate increased gradually with each 10-year age group, from 0.2% in the youngest subpopulation (<20 years) to 34.2% in the eldest group (80+ years) (RR = 66.96; 95% CI = 52.35–85.65). A higher proportion of pregnant women were hospitalized than women who were not pregnant (2.5% vs. 1.1%).

### 3.2. Chronic Comorbidities as Risk Factors

The proportion of individuals with comorbidities among hospital admissions was higher than in those who did not require hospitalization (Table 1), including hypertension (54.1% vs. 14%), depression (38.3% vs. 21.7%), diabetes (35.8% vs. 8.3%), osteoarthritis (27.3% vs. 6.6%), CHD (26.9% vs. 4.4%), cancer (26.8% vs. 9.9%), CKD (23.5% vs. 3.0%), asthma (20.1% vs. 13.1%), substance use disorder (13.7% vs. 4.3%), problematic alcohol use (13.6% vs. 4.1%), heart failure (11.7% vs. 1.1%), COPD (11.5% vs. 1.5%), IDU (10% vs. 3.8%) and immunosuppression (6.0% vs. 2.2%).

In the adjusted multivariable Poisson regression model (Table 2; Figure 1), age (*p*-trend < 0.0001 across age groups with increasing risk with older age [aRR 30–39 years = 3.06; 95% CI = 2.32–4.03, to aRR 80+ years = 43.68; 95% CI = 33.41–57.10 compared to 20–29 years-old]), male sex (aRR = 1.27; 95% CI = 1.17–1.37), asthma (aRR = 1.15; 95% CI = 1.04–1.26), cancer (1.19; 95% CI = 1.09–1.29), CKD (RR = 1.32; 95% CI = 1.19–1.47), diabetes (treated without insulin aRR = 1.13; 95% CI = 1.03–1.25, requiring insulin aRR = 5.05; 95% CI = 4.43–5.76), hypertension (aRR = 1.19; 95% CI = 1.08–1.31), immunosuppression(1.30; 95% CI = 1.10–1.54), IDU (aRR = 2.51; 95% CI = 2.14–2.95), IDD (aRR = 1.67; 95% CI = 1.05–2.66), problematic alcohol use (aRR = 1.63; 95% CI = 1.43–1.85) and SZP (aRR = 1.49; 95% CI = 1.23–1.82) were associated with increased hospitalization risk. The analysis by time period (Appendix A) did not show any remarkable differences in risk factors.

Among women of reproductive age (Table 3), pregnancy (aRR = 3.05; 95% CI = 1.86–5.07), older age(*p*-trend < 0.0001 across age groups; RR = 2.32; 95% CI = 1.54–3.48 comparing 40–49 to 20–29 years-old groups), asthma (aRR = 1.80; 95% CI = 1.29–2.52), diabetes (treated without insulin aRR = 2.39; 95% CI = 1.46–3.89, requiring insulin aRR = 31.89; 95% CI = 16.78–60.60), hypertension (aRR = 2.02; 95% CI = 1.29–3.16), IDU (aRR = 3.97; 95% CI = 2.44–6.43), and problematic alcohol use (aRR = 3.05; 95% CI = 1.86–5.02), were factors significantly associated with higher risk of hospital admission.

In the analysis stratified by age (Table 4), diabetes, and problematic alcohol use were associated with hospitalization across all age groups, with higher risk among younger groups that decreased with older age. Male sex, cancer and hypertension were associated with higher risk of hospital admission among those over 40 years of age. The analysis restricted to hospitalizations lasting more than two days (Appendix A) did not show different findings compared with the overall analysis.

## 4. Discussion

In this large population-based analysis of all COVID-19 cases in BC, we identified several risk factors for COVID-19 hospitalization. Older age was the strongest predictor of hospital admission, with risk increasing more than 40 fold for the oldest group compared to the reference (20–29 years-old). In addition to well-characterized co-morbidities, we found that injection drug use [5], problematic alcohol use, schizophrenia and psychotic disorders [16] as well as intellectual and developmental disability were independently associated with higher risk of hospitalization. These findings have important implications for vaccination programs. For instance, this evidence was used to identify priority groups, informing the COVID-19 vaccination program in BC [17], with the ultimate aim of preventing infection and severe outcomes as well as reducing hospital burden.

Our analysis confirms findings from other studies evaluating risk factors for severe COVID-19 outcomes, although ours is one of the few population-based analyses (i.e., includes all COVID-19 diagnoses in a jurisdiction). Most evaluations have also focused on in-hospital mortality, rather than morbidity. In addition to older age and male sex, a wide range of co-morbidities were associated with a higher risk of hospitalization, reflecting similar findings from other studies [7,18,19,20,21,22,23]. These comorbidities included asthma, chronic kidney disease, diabetes, cancer, immunosuppression and substance use. Relative associations between most co-morbidities and hospitalization risk were stronger at younger ages, highlighting the overall low absolute risk of hospitalization among younger people without pre-existing comorbidities. However, hospitalization risk increased with older age in the overall population, and the highest absolute risk was observed in people of older ages with co-morbidities. Several biological studies have identified sex and age differences in biological pathways related to SARS-CoV-2 infection and support our findings [24,25,26].

As in our study, pregnancy has been previously identified as a potential risk factor for ICU admission [27,28] and severe disease [29]. However, most other studies have been limited to pregnant women who were already hospitalized (including for non-COVID-19 reasons such as childbirth) [27,30]. This finding could be in part the result of a lower clinical threshold for hospitalization of pregnant patients.

In our study, insulin-dependent diabetes was associated with higher risk of hospitalization, particularly among individuals younger than 40 years. To our knowledge, this is the first report observing this phenomenon, although insulin use and increased risk of COVID-19-related death was described earlier in the literature. Further research is needed to better characterize this finding.

Our study also highlights the intersection between the two ongoing public health emergencies in BC: the COVID-19 and the opioid overdose epidemics. The COVID-19 pandemic has exacerbated the pre-existing opioid epidemic in several ways, including disruption of harm reduction services [31], with BC experiencing a record high number of illicit drug toxicity deaths in 2020 [32]. This is the first study investigating the impact of COVID-19 on people who inject drugs. Our findings suggest that individuals at high risk of overdose, as indicated by IDU, are also at higher risk of COVID-19 hospitalization. IDU was the third strongest predictor of hospitalization in our analysis (following older age and insulin-dependent diabetes). Similarly, problematic alcohol use, schizophrenia and psychotic disorders were also associated with higher risk of hospitalization. These findings may highlight the syndemic of substance use, mental illness and COVID-19. Underlying social conditions (e.g., unstable housing, lower socioeconomic status) and many co-occurring co-morbidities may have exacerbated the effect of COVID-19 infection among these individuals. Prioritization of vaccination for this population group could reduce disparities and decrease risk of hospitalization.

Our analysis had several limitations. We relied on administrative data to identify patient characteristics and co-morbidities; this may have led to some level of misclassification. Similarly, for the same reason, it is not possible to evaluate clinical severity of the event leading to hospital admission. Further, we did not have information on socioeconomic status, race/ethnicity, obesity, and stage of pregnancy (trimester). Moreover, given that the evaluation of the COVID-19 status depends on the BC diagnostic testing guidelines (varying over time to focus on symptom-based assessment since 21 April 2020) [33], selective ascertainment of symptomatic cases is expected, resulting in exclusion of asymptomatic cases [34].

## 5. Conclusions

In conclusion, older age, male sex, pregnancy, and various comorbidities and health-conditions, including substance use, were associated with higher risk of hospital admission in this population-based analysis. These findings have informed the COVID-19 vaccination program rollout in BC and can be useful for informing the prioritization of vaccination in other jurisdictions to prevent infection and severe outcomes [17]. In addition, these findings could also guide healthcare providers in the monitoring of individual patients at higher risk of severe outcomes. Finally, the evidence shows the need for further characterizing syndemics of substance use, mental illness and COVID-19.

## Figures and Tables

**Figure 1 viruses-13-02196-f001:**
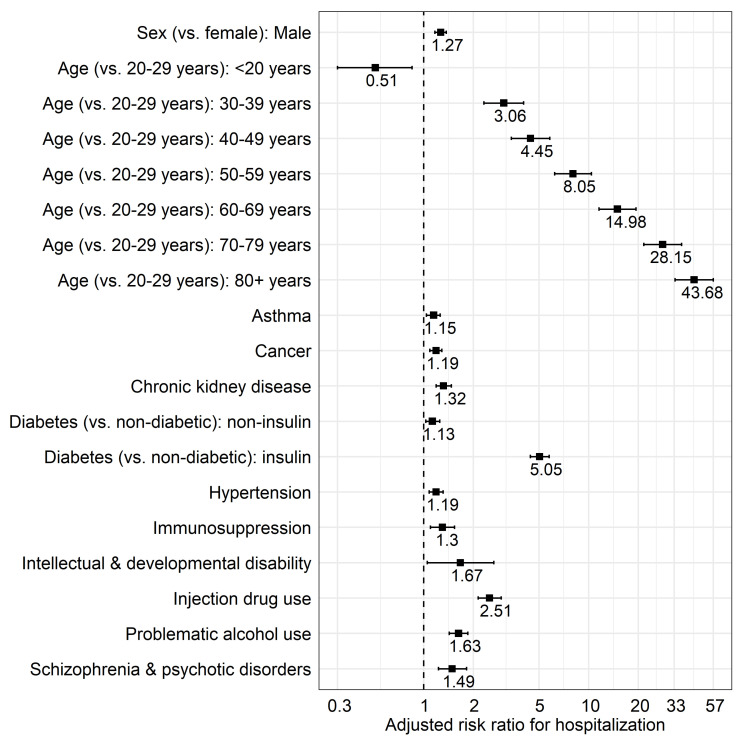
Multivariable model for factors associated with COVID-19–related hospitalization in British Columbia.

**Table 1 viruses-13-02196-t001:** Distribution of characteristics in the BC COVID-19 cohort (confirmed cases, *n* = 56,874), according to hospitalization status.

Variable	Category	Total * *n* = 56,874	Non-Hospitalized ⁰ *n* = 54,576	Hospitalized ⁰ *n* = 2298	Hospitalized Row %	Crude Incidence Rate Ratio (95% Conf. Interv.)	*p* ^†^
Sex	Female	27,769 (48.8%)	26,816 (49.1%)	953 (41.5%)	3.4%	Reference	<0.0001
	Male	29,105 (51.2%)	27,760 (50.9%)	1345 (58.5%)	4.6%	1.35 (1.24–1.46)	
Age (years)	N/A	36 (28) ⁱ	35 (26) ⁱ	66 (25) ⁱ	-	1.07 (1.06–1.07)	<0.0001
Age group ^‡^	<20 years	7683 (13.5%)	7665 (14.0%)	18 (0.8%)	0.2%	0.45 (0.27–0.77)	0.0031
	20–29 years	13,490 (23.7%)	13,421 (24.6%)	69 (3.0%)	0.5%	Reference	-
	30–39 years	10,681 (18.8%)	10,507 (19.3%)	174 (7.6%)	1.6%	3.18 (2.41–4.20)	<0.0001
	40–49 years	8818 (15.8%)	8604 (15.8%)	214 (9.3%)	2.4%	4.74 (3.62–6.22)	<0.0001
	50–59 years	7565 (13.3%)	7201 (13.2%)	364 (15.8%)	4.8%	9.41 (7.28–12.15)	<0.0001
	60–69 years	4793 (8.4%)	4330 (7.9%)	463 (20.1%)	9.7%	18.89 (14.70–24.27)	<0.0001
	70–79 years	2425 (4.3%)	1915 (3.5%)	510 (22.2%)	21.0%	41.12 (32.10–52.67)	<0.0001
	80+ years	1419 (2.5%)	933 (1.7%)	486 (21.1%)	34.2%	66.96 (52.35–85.65)	<0.0001
Pregnant (female population, ages 15 to 49; *n* = 17,693)	No	16,638 (94.0%)	16,457 (94.1%)	181 (86.6%)	1.1%	Reference	-
	Yes	398 (2.2%)	388 (2.2%)	10 (4.8%)	2.5%	2.31 (1.23–4.33)	0.0091
	Unknown	657 (3.7%)	639 (3.7%)	18 (8.6%)	2.7%	2.52 (1.56–4.06)	0.0002
Angina	No	55,915 (98.3%)	53,850 (98.7%)	2065 (89.9%)	3.7%	Reference	<0.0001
	Yes	959 (1.7%)	726 (1.3%)	233 (10.1%)	24.3%	6.58 (5.84–7.41)	
Chronic heart disease ^1^	No	53,840 (94.7%)	52,161 (95.6%)	1679 (73.1%)	3.1%	Reference	<0.0001
	Yes	3034 (5.3%)	2415 (4.4%)	619 (26.9%)	20.4%	6.54 (6.01–7.12)	
Heart failure	No	56,003 (98.5%)	53,975 (98.9%)	2028 (88.3%)	3.6%	Reference	<0.0001
	Yes	871 (1.5%)	601 (1.1%)	270 (11.7%)	31.0%	8.56 (7.68–8.54)	
Hypertension	No	48,009 (84.4%)	46,955 (86.0%)	1054 (45.9%)	2.2%	Reference	<0.0001
	Yes	8865 (15.6%)	7621 (14.0%)	1244 (54.1%)	14.0%	6.39 (5.91–6.92)	
Ischemic heart disease	No	54,139 (95.2%)	52,378 (96.0%)	1761 (76.6%)	3.3%	Reference	<0.0001
	Yes	2735 (4.8%)	2198 (4.0%)	537 (23.4%)	19.6%	6.04 (5.52–6.59)	
Myocardial infarct	No	56,273 (98.9%)	54,109 (99.1%)	2164 (94.2%)	3.8%	Reference	<0.0001
	Yes	601 (1.1%)	467 (0.9%)	134 (5.8%)	22.3%	5.80 (4.97–6.77)	
Immunosuppression ^2^	No	55,541 (97.7%)	53,380 (97.8%)	2161 (94.0%)	3.9%	Reference	<0.0001
	Yes	1333 (2.3%)	1196 (2.2%)	137 (6.0%)	10.3%	2.64 (2.24–3.11)	
Rheumatoid arthritis	No	56,198 (98.8%)	53,998 (98.9%)	2200 (95.7%)	3.9%	Reference	<0.0001
	Yes	676 (1.2%)	578 (1.1%)	98 (4.3%)	14.5%	3.70 (3.07–4.47)	
Depression	No	44,541 (78.3%)	43,124 (79.0%)	1417 (61.7%)	3.2%	Reference	<0.0001
	Yes	12,333 (21.7%)	11,452 (21.0%)	881 (38.3%)	7.1%	2.25 (2.07–2.44)	
Intellectual and developmental disability ^3^	No	56,444 (99.2%)	54,177 (99.3%)	2267 (98.7%)	4.0%	Reference	0.0008
	Yes	430 (0.8%)	399 (0.7%)	31 (1.3%)	7.2%	1.79 (1.28–2.53)	
Mood and anxiety disorders	No	41,590 (73.1%)	40,322 (73.9%)	1268 (55.2%)	3.0%	Reference	<0.0001
	Yes	15,284 (26.9%)	14,254 (26.1%)	1030 (44.8%)	6.7%	2.21 (2.04–2.39)	
Schizophrenia and psychotic disorders	No	56,112 (98.7%)	53,927 (98.8%)	2185 (95.1%)	3.9%	Reference	<0.0001
	Yes	762 (1.3%)	649 (1.2%)	113 (4.9%)	14.8%	3.81 (3.20–4.53)	
Diabetes	No	51,487 (90.5%)	50,011 (91.6%)	1476 (64.2%)	2.9%	Reference	-
	Non-insulin	5141 (9.1%)	4541 (8.3%)	600 (26.1%)	11.7%	4.07 (3.72–4.46)	<0.0001
	Insulin ^5^	246 (0.4%)	24 (0.0%)	222 (9.7)	90.2%	31.48 (29.50–33.59)	<0.0001
Gout	No	55,707 (97.9%)	53,622 (98.3%)	2085 (90.7%)	3.7%	Reference	<0.0001
	Yes	1167 (2.1%)	964 (1.7%)	213 (9.3%)	18.3%	4.88 (4.29–5.55)	
Chronic kidney disease ^4^	No	54,686 (96.2%)	52,928 (97.0%)	1758 (76.5%)	3.2%	Reference	<0.0001
	Yes	2188 (3.8%)	1648 (3.0%)	540 (23.5%)	24.7%	7.68 (7.04–8.37)	
Osteoarthritis	No	52,645 (92.6%)	50,975 (93.4%)	1670 (72.7%)	3.2%	Reference	<0.0001
	Yes	4229 (7.4%)	3601 (6.6%)	628 (27.3%)	14.8%	4.68 (4.29–5.10)	
Osteoporosis	No	55,831 (98.2%)	53,771 (98.5%)	2060 (89.6%)	3.7%	Reference	<0.0001
	Yes	1043 (1.8%)	805 (1.5%)	238 (10.4%)	22.8%	6.18 (5.49–6.97)	
Alzheimer/dementia	No	56,624 (99.6%)	54,411 (99.7%)	2213 (93.3%)	3.9%	Reference	<0.0001
	Yes	250 (0.4%)	165 (0.3%)	85 (3.7%)	34.0%	8.70 (7.28–10.39)	
Epilepsy	No	56,465 (99.3%)	54,216 (99.3%)	2249 (97.9%)	4.0%	Reference	<0.0001
	Yes	409 (0.7%)	360 (0.7%)	49 (2.1%)	12.0%	3.01 (2.31–3.92)	
Parkinsonism	No	56,806 (99.9%)	54,534 (99.9%)	2272 (98.9%)	4.0%	Reference	<0.0001
	Yes	68 (0.1%)	42 (0.1%)	26 (1.1%)	38.2%	9.56 (7.05–12.97)	
Stroke (combined) ^5^	No	56,571 (99.5%)	54,351 (99.6%)	2220 (96.6%)	3.9%	Reference	<0.0001
	Yes	303 (0.5%)	225 (0.4%)	78 (3.4%)	25.7%	6.56 (5.39–7.98)	
Stroke (hemorrhagic)	No	56,817 (99.9%)	54,528 (99.9%)	2289 (99.6%)	4.0%	Reference	<0.0001
	Yes	57 (0.1%)	48 (0.1%)	9 (0.4%)	15.8%	3.92 (2.15–7.15)	
Stroke (ischemic)	No	56,705 (99.7%)	54,453(99.8%)	2252 (98.0%)	4.0%	Reference	<0.0001
	Yes	169 (0.3%)	123 (0.2%)	46 (2.0%)	27.2%	6.85 (5.34–8.80)	
Transitory ischemic attack	No	56,773 (99.8%)	54,504 (99.9%)	2269 (98.7%)	4.0%	Reference	<0.0001
	Yes	101 (0.2%)	72 (0.1%)	29 (1.3%)	28.7%	7.18 (5.27–9.79)	
Asthma	No	49,270 (86.6%)	47,435 (86.9%)	1835 (79.9%)	3.7%	Reference	<0.0001
	Yes	7604 (13.4%)	7141 (13.1%)	463 (20.1%)	6.1%	1.63 (1.48–1.81)	
Chronic obstructive pulmonary disease	No	55,800 (98.1%)	53,767 (98.5%)	2033 (88.5%)	3.6%	Reference	<0.0001
	Yes	1074 (1.9%)	809 (1.5%)	265 (11.5%)	24.7%	6.77 (6.05–7.58)	
Injection drug use ^6^	No	54,574 (96.0%)	52,506 (96.2%)	2068 (90.0%)	3.8%	Reference	<0.0001
	Yes	2300 (4.0%)	2070 (3.8%)	230 (10.0%)	10.0%	2.64 (2.32–3.00)	
Problematic alcohol use ^7^	No	54,322 (95.5%)	52,336 (95.9%)	1986 (86.4%)	3.7%	Reference	<0.0001
	Yes	2552 (4.5%)	2240 (4.1%)	312 (13.6%)	12.2%	3.34 (2.99–3.74)	
Cancer ^8^	No	50,865 (89.4%)	49,182 (90.1%)	1683 (73.2%)	3.3%	Reference	<0.0001
	Yes	6009 (10.6%)	5394 (9.9%)	615 (26.8%)	10.2%	3.09 (2.83–3.38)	
Cirrhosis ^7^	No	56,635 (99.6%)	54,402 (99.7%)	2233 (97.2%)	3.9%	Reference	<0.0001
	Yes	239 (0.4%)	174 (0.3%)	65 (2.8%)	27.2%	6.90 (5.58–8.52)	

***** Number of participants (proportion within variable). ⁰ Number of participants (proportion within hospitalization status subgroup). ^†^ Bivariate Poisson regression with robust error variance (Wald’s test). ^‡^
*p*-trend < 0.0001 (age groups assessed as pseudo-continuous values). ⁱ Median (interquartile range). ^1^ Combination of acute myocardial infarct, chronic heart failure and ischemic heart disease. ^2^
*Sundaram ME* et al., 2021 [12]. ^3^ Based on ICD-9/ICD-10 codes from: Manitoba Centre for Health Policy [13]. ^4^ Assessed via “renal disease” ICD-9/ICD-10 codes from group 14 of Elixhauser Comorbidity Score in DAD, MSP and NACRS records. ^5^ Combination of hemorrhagic stroke, stroke (hospitalized), transitory ischemic attack (hospitalized) and ischemic stroke. ^6^ Janjua, N.Z. et al., 2018 [14]. ^7^ Janjua, N.Z. et al., 2016 [15]. ^8^ Assessed via “lymphoma”, “metastatic cancer” and “solid tumor without metastasis” ICD-9/ICD-10 codes from groups 18, 19 and 20 of Elixhauser Comorbidity Score in DAD, MSP and NACRS records.

**Table 2 viruses-13-02196-t002:** Factors associated with hospitalization status in multivariable Poisson regression analysis with robust error variance among confirmed cases, BC COVID-19 Cohort.

Variable	Category	26 January 2020–15 January 2021 (*n* = 56,874; Hospitalized = 2298)	
		aRR (95% CI) *	*p* ^†^
Sex (vs. female)	Male	1.27 (1.17–1.37)	<0.0001
Age ^‡^ (Reference group: 20–29 years)	<20 years	0.51 (0.30–0.85)	0.0103
	30–39 years	3.06 (2.32–4.03)	<0.0001
	40–49 years	4.45 (3.40–5.82)	<0.0001
	50–59 years	8.05 (6.22–10.41)	<0.0001
	60–69 years	14.98 (11.58–19.37)	<0.0001
	70–79 years	28.15 (21.64–36.61)	<0.0001
	80+ years	43.68 (33.41–57.10)	<0.0001
Asthma		1.15 (1.04–1.26)	0.0049
Cancer ^1^		1.19 (1.09–1.29)	0.0001
Chronic kidney disease ^2^		1.32 (1.19–1.47)	<0.0001
Diabetes (vs. non-diabetic)	Non-insulin	1.13 (1.03–1.25)	0.0112
	Insulin ^3^	5.05 (4.43–5.76)	<0.0001
Hypertension		1.19 (1.08–1.31)	0.0007
Immunosuppression ^4^		1.30 (1.10–1.54)	0.0019
Injection drug use ^5^		2.51 (2.14–2.95)	<0.0001
Intellectual and developmental disability ^6^		1.67 (1.05–2.66)	0.0307
Problematic alcohol use ^7^		1.63 (1.43–1.85)	<0.0001
Schizophrenia and psychotic disorders		1.49 (1.23–1.82)	<0.0001

* Incidence rate ratios adjusted for the variables present in the table. ^†^ Wald’s test. ^‡^
*p*-trend > 0.0001 (age groups assessed as pseudo-continuous values) ^1^ Assessed via “lymphoma”, “metastatic cancer” and “solid tumor without metastasis” ICD-9/ICD-10 codes from groups 18, 19 and 20 of Elixhauser Comorbidity Score in DAD, MSP and NACRS records. ^2^ Assessed via “renal disease” ICD-9/ICD-10 codes from group 14 of Elixhauser Comorbidity Score in DAD, MSP and NACRS records. ^3^ Any type; includes concomitant treatment with antihyperglycemic agents. ^4^ Sundaram, M.E. et al., 2021 [12]. ^5^ Janjua, N.Z.et al., 2018 [14]. ^6^ Based on ICD-9/ICD-10 codes from Manitoba Centre for Health Policy [13]. ^7^ Janjua, N.Z. et al., 2016 [15].

**Table 3 viruses-13-02196-t003:** Factors associated with hospitalization status in multivariable Poisson regression analysis with robust error variance among women of reproductive age (15–49 years-old), BC COVID-19 Cohort ^‡^.

Variable	Category	26 January 2020–15 January 2021 (*n* = 17,036; Hospitalized = 191)	
		aRR (95% CI) *	*p* ^†^
Age ** (Reference group: 20–29 years)	<20 years	0.24 (0.06–0.95)	0.0424
	30–39 years	1.99 (1.35–2.94)	0.0005
	40–49 years	2.32 (1.54–3.48)	<0.0001
Asthma		1.80 (1.29–2.52)	0.0005
Diabetes (vs. non-diabetic)	Non-insulin	2.39 (1.46–3.89)	0.0005
	Insulin ^1^	31.89 (16.78–60.60)	<0.0001
Hypertension		2.02 (1.29–3.16)	0.0020
Injection drug use ^2^		3.97 (2.44–6.43)	<0.0001
Pregnancy		2.69 (1.42–5.07)	0.0023
Problematic alcohol use ^3^		3.05 (1.86–5.02)	<0.0001

^‡^ 657 observations with missing pregnancy status were removed from original sample (*n* = 17,693). * Incidence rate ratios adjusted for the variables present in the table. ** *p*-trend > 0.0001 (age groups assessed as pseudo-continuous values). ^†^ Wald’s test. ^1^ Any type; includes concomitant treatment with antihyperglycemic agents. ^2^ Janjua, N.Z. et al., 2018 [14]. ^3^ Janjua, N.Z. et al., 2016 [15].

**Table 4 viruses-13-02196-t004:** Factors associated with hospitalization status in multivariable Poisson regression analysis with robust error variance among confirmed cases, BC COVID-19 Cohort, stratified by age group.

Variable		Age Group							
		All Age Groups ⁰ (*n* = 56,874; Hospitalized = 2298)		<40 Years (*n* = 31,854; Hospitalized = 261)		40–59 Years (*n* = 16,383; Hospitalized = 578)		60+ Years (*n* = 8637; Hospitalized = 1459)	
		aRR (95% CI) *	*p* †	aRR (95% CI) *	*p* ^†^	aRR (95% CI) *	*p* ^†^	aRR (95% CI) *	*p* ^†^
Sex (male vs. female)		1.27 (1.17–1.37)	<0.0001	0.99 (0.77–1.27)	0.93	1.46 (1.24–1.72)	<0.0001	1.26 (1.15–1.38)	<0.0001
Asthma		1.15 (1.04–1.26)	0.0049	1.10 (0.78–1.54)	0.59	1.41 (1.16–1.71)	0.0005	1.10 (0.98–1.23)	0.09
Cancer ^1^		1.19 (1.09–1.29)	0.0001	1.47 (0.99–2.17)	0.054	1.24 (1.00–1.54)	0.0459	1.31 (1.19–1.44)	<0.0001
Chronic kidney disease ^2^		1.32 (1.19–1.47)	<0.0001	2.68 (1.22–5.90)	0.0141	1.32 (0.97–1.80)	0.08	1.70 (1.53–1.88)	<0.0001
Diabetes (vs. non-diabetic)	Non-insulin	1.13 (1.03–1.25)	0.0112	2.03 (0.98–4.17)	0.055	1.62 (1.31–1.99)	<0.0001	1.01 (0.91–1.12)	0.87
	Insulin ^3^	5.05 (4.43–5.76)	<0.0001	20.29 (4.69–87.90)	<0.0001	19.30 (14.66–25.40)	<0.0001	3.94 (3.52–4.41)	<0.0001
Hypertension		1.19 (1.08–1.31)	0.0007	1.87 (0.98–3.57)	0.057	1.46 (1.21–1.75)	<0.0001	1.31 (1.18–1.46)	<0.0001
Immunosuppression ^4^		1.30 (1.10–1.54)	0.0019	2.56 (1.67–3.94)	<0.0001	1.30 (0.95–1.76)	0.10	1.14 (0.93–1.38)	0. 21
Injection drug use ^5^		2.51 (2.14–2.95)	<0.0001	2.95 (1.78–4.90)	<0.0001	2.55 (1.97–3.30)	<0.0001	1.01 (0.76–1.32)	0.97
Intellectual and developmental disability ^6^		1.67 (1.05–2.66)	0.0307	1.17 (0.56–2.42)	0.68	4.28 (2.29–8.01)	<0.0001	1.21 (0.71–2.04)	0.49
Problematic alcohol use ^7^		1.63 (1.43–1.85)	<0.0001	2.84 (1.72–4.72)	<0.0001	1.62 (1.26–2.10)	0.0002	1.53 (1.30–1.79)	<0.0001
Schizophrenia and psychotic disorders		1.49 (1.23–1.82)	<0.0001	2.33 (1.40–3.87)	0.0012	1.19 (0.80–1.76)	0.38	1.56 (1.21–2.00)	0.0005

* Incidence rate ratios adjusted for the variables present in the table. ⁰ Adjusted for also for age (as categorical groups from Table 2; not shown). ^†^ Wald’s test. ^1^ Assessed via “lymphoma”, “metastatic cancer” and “solid tumor without metastasis” ICD-9/ICD-10 codes from groups 18, 19 and 20 of Elixhauser Comorbidity Score in DAD, MSP and NACRS records. ^2^ Assessed via “renal disease” ICD-9/ICD-10 codes from group 14 of Elixhauser Comorbidity Score in DAD, MSP and NACRS records. ^3^ Any type; includes concomitant treatment with antihyperglycemic agents. ^4^ Sundaram, M.E. et al., 2021 [12]. ^5^ Janjua, N.Z. et al., 2018 [14]. ^6^ Based on ICD-9/ICD-10 codes from Manitoba Centre for Health Policy [13]. ^7^ Janjua, N.Z. et al., 2016 [15].

## Data Availability

The study is based on data contained in various provincial registries and databases. Access to data could be requested through the BC Centre for Disease Control Institutional Data Access for researchers who meet the criteria for access to confidential data. Requests for the data may be sent to datarequest@bccdc.ca.

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
