# Peer review of "Mental Health and Substance Use Associated with Hospitalization among People with COVID-19: A Population-Based Cohort Study"

_viruses, 2021, doi:10.3390/v13112196_

Round 1

Reviewer 1 Report

Some improvements are required:

  1. The authors employ a statistical approach. However, there are various other approaches to tackle this problem. Please, acknowledge those approaches in the introduction.
  2. The writing is sloppy. Moreover, the paper needs cosmetic work. As it is, it seems that the work was completed in a Rush.
  3. The contributions are very small. I recommend that the authors include the phrase "Note on ..." in the title of the paper. In that way, the readers will know that the work is a rather short contribution.
  4. A more thorough description of the statistical methods is required. As it is, the scientists in statistics will be able to evaluate the methodology employed.

Author Response

Response to Reviewer 1 Comments

  1. The authors employ a statistical approach. However, there are various other approaches to tackle this problem. Please, acknowledge those approaches in the introduction.

We thank the review for comment. For the current analysis, we used a cohort study design and used analyses techniques appropriate to the cohort design (multivariable Poisson regression models with robust error variance).  There could be other statistical approaches such as logistic regression to compute of odds ratio, and survival analysis to compute hazards ratio. We did not perform logistic regression, as odds ratios are not considered appropriate for cohort studies (estimation of odds ratios is possible and valid in cohort designs, but the odds ratio is often not the estimate that is desired when information about person-time at risk is available; also, even though the odds ratio for rare outcomes will approximately estimate the risk ratio in the cohort design, it makes more sense to compute the risk ratio directly [Bhopal RS. Epidemiological study design and principles of data analysis: An integrated suite of methods. In: Concepts of Epidemiology: Integrating the ideas, theories, principles and methods of epidemiology. 2nd edition ed. Oxford University Press; 2008. p. 315-388]). Given that follow-up time from infection to hospitalization is short and ascertainment is complete, we did not use survival analyses.

  1. The writing is sloppy. Moreover, the paper needs cosmetic work. As it is, it seems that the work was completed in a Rush.

We thank the reviewer for this observation.  We have proofread the document and updated the language. We hope the current writing is in a more acceptable format. 

  1. The contributions are very small. I recommend that the authors include the phrase "Note on ..." in the title of the paper. In that way, the readers will know that the work is a rather short contribution.

We respectfully disagree with the review. This is one of the few population based studies assessing risk factors for hospitalization. In addition, we have evaluated and identified novel risk factors such as role of mental health and substance use in addition to providing data on insulin-dependent diabetes. We also assessed risk factors among women and found pregnancy as one of the hospitalization risks. These are major contributions to the knowledge in this area.

  1. A more thorough description of the statistical methods is required. As it is, the scientists in statistics will be able to evaluate the methodology employed.

We have revised and updated the “Statistical analysis” section to provide more details on the model building process.

Reviewer 2 Report

This is a retrospective study to identify risk factors associated with hospital admissions among patients with Covid-19. Older age, male sex, pregnancy, chronic co-morbidities and substance abuse are associated with increased risk of hospitalisation.

Please, specify the age in the abstract, instead of "older age" as a risk factor for hospital admission. 

Moreover, among pregnant women, it would be suitable to specify if there is an increased risk for hospital admission according to the trimester of pregnancy of Covid-19 infection (first, second or third). 

Author Response

Response to Reviewer 2 Comments

  1. Please, specify the age in the abstract, instead of "older age" as a risk factor for hospital admission.

We thank the review for comment. However, as shown in the manuscript, we found an increase in COVID-19 hospitalization risk with increasing age after age 30 years, when using 20-29 years as reference, or when using age a continue variable, so it was not possible to identify a specific age at which the risk was greater.

  1. Moreover, among pregnant women, it would be suitable to specify if there is an increased risk for hospital admission according to the trimester of pregnancy of Covid-19 infection (first, second or third).

​We agree with the reviewer. Unfortunately, we do not have access to data to ascertain the trimester of pregnancy. We have included this as a limitation in the "Discussion" section.

Round 2

Reviewer 1 Report

I believe that the paper is still a modest contribution. Little have the authors done to improve it and they aren't willing to do it. I will accept this submission in view that the results may be useful, in spite of being preliminary.